# Mapping of Danish Pharmacy Technician Students’ Third-Year Projects in a Year with the COVID-19 Pandemic

**DOI:** 10.3390/pharmacy10010033

**Published:** 2022-02-17

**Authors:** Bjarke Abrahamsen, Rikke Nørgaard Hansen, Mette Skjøtt, Ditte Sloth-Lisbjerg, Charlotte Verner Rossing

**Affiliations:** 1Department of Research & Development, The Danish College of Pharmacy Practice, Milnersvej 42, 3400 Hillerød, Denmark; rnh@pharmakon.dk (R.N.H.); cr@pharmakon.dk (C.V.R.); 2Department of Education, The Danish College of Pharmacy Practice, Milnersvej 42, 3400 Hillerød, Denmark; ms@pharmakon.dk (M.S.); dsl@pharmakon.dk (D.S.-L.)

**Keywords:** COVID-19, pharmacy education, pharmacy technician, pharmaconomist, Denmark

## Abstract

To graduate, pharmacy technician students write a project in their third year. They choose between six elective courses, and work with a subject related to their education and everyday practice at community or hospital pharmacies. In this article, we report the mapping of third-year project themes and provide an overview of the challenges that COVID-19 pandemic restrictions have had on completing the projects. On the basis of all project titles, a list of themes was generated and described before all projects were allocated to one of the themes. Challenges experienced due to the COVID-19 pandemic were investigated from an analytical workshop where supervisors discussed their experience with supervising students throughout the completion of the projects. In total, 140 projects were included and thematised into eight themes: advanced pharmacy services, digital patient support, organisation and collaboration, handling of medicine, automated dose dispensing, medication counselling in community pharmacy, hospital pharmacy, and others, covering all six elective courses. The COVID-19 pandemic affected students’ possibilities to collect data from either physical interviews or observations. The challenges prompted both constructive and creative discussions between students and supervisors to find ways to complete the projects, and required flexibility from all those involved: students, supervisors, community pharmacies, and hospital pharmacies. In conclusion, all students managed to complete their third-year project at a similar level of achievement statistically compared to average grades for the previous six years (2016–2020).

## 1. Introduction

In Denmark, pharmacy technician students complete their three-year education with a third-year project. From a choice of six election courses (Table 1), students work with a relevant subject to their practice in either a community pharmacy or hospital pharmacy. The COVID-19 pandemic caused significant changes to the way that community and hospital pharmacies could provide health-related services [1,2,3]. During the COVID-19 pandemic, Danish community pharmacies have remained a place with frontline community health professionals who can provide patients with face-to-face advice without patients having to book an appointment. At hospital pharmacies, some staff have been part of the emergency response and have had to prepare information on how to deal with the situation. The lockdown entailed many changes at both community pharmacies and hospital pharmacies, e.g., changes to the physical surroundings, work schedules, and emergency arrangements with staff associations. The possibility for both pre- and postgraduate education was also deeply affected, with all physical contact replaced by online teaching. For both community pharmacies and hospital pharmacies, this meant long working hours and increased work pressure to adapt to the constant changes [3]. 

This study provides a mapping of pharmacy technician students’ third-year projects and uncovers the challenges related to completing their projects caused by the COVID-19 pandemic. 

### Danish Pharmacy Technician Students and Elective Course with Project

The Danish pharmacy technician programme is a three-year education programme equivalent to 180 European Credit Transfer System (ECTS) points. The academic part takes place at the Danish College of Pharmacy Practice and corresponds to 85 ECTS points. The students spend a total of 23 weeks taking eight courses at the college. The practical part of the education corresponds to 95 ECTS points and takes place at a community pharmacy, where the students are employed. 

As part of the students’ final year, they carry out a project equivalent to 8 ECTS points. The students can choose between six different courses related to pharmacy practice, as outlined in Table 1.

Each student defines a research question, aim, and objectives for the project. Throughout their time of study, students work at their designated community pharmacy. Students who choose to do their third-year project at a hospital pharmacy still work at their designated community pharmacy, but have a total of two weeks allocated to a hospital pharmacy. Students have a project supervisor from the Danish College of Pharmacy Practice who is also teacher in the elective course. Students complete their third-year project with a report that is assessed by external reviewers and graded according to a 7-point scale (−3, 0, 02, 4, 7, 10 and 12), where grades from 02 to 12 mean that students passed. The format for the third-year project reports follows internal guidelines with requirements for a paper consisting of 36,000–48,000 characters.

## 2. Materials and Methods

To generate themes, project titles were extracted from students’ third-year project reports and listed randomly before being distributed to five researchers. Each researcher generated a list of overall themes identified by reading the project titles, and an overall list of themes was generated through consensus decision making by all five researchers. The list of themes was further presented and discussed with two project supervisors. Lastly, all projects were allocated to one of the final themes. A brief description of each theme and the number of allocated projects was reported. Please see Figure 1 for an overview of the workflow for the process.

Descriptive data included the total number of projects, the number of projects with consent to be used for this review, the number of projects per theme, and the number of projects associated with the identified subthemes. Each subtheme was briefly described. Analysis of differences for the average grades for the last six years (2016–2021) was performed using ANOVA, and an one-sided t-test was used to compare data for 2021 with 2020. Microsoft Excel 365 was used for descriptive and statistical analysis.

Mapping themes, and challenges and limitations of students’ third-year projects due to the COVID-19 pandemic were discussed and explored by researchers and two project supervisors at an analytical workshop. Two themes representing the most dominant projects from community pharmacies and hospital pharmacies were chosen for further description. Statements from supervisors concerning challenges and limitations experienced with students due to the COVID-19 pandemic were registered and summarised by researchers. Lastly, the project supervisors validated the summary to clarify any misunderstandings. Please see Figure 1 for an overview of the process.

## 3. Results

A total of 178 projects were mapped, with 38 projects being excluded because they did not have consent for use in research. The 140 included projects were each allocated to one of the final eight themes. The allocation of each included project is shown in Table 2.

Medication counselling in community pharmacy (n = 52) was further differentiated into patient subgroups: diabetes (n = 10), osteoporosis (n = 6), depression (n = 2), users of inhalation medicine (n = 2), blood pressure (n = 4), NSAID (n = 5), contraception (n = 12), nicotine substitution (n = 4), reflux (n = 1), users of methotrexate (n = 1), users of prednisolone (n = 1), vaginal thrush (n = 1), infertility (n = 1), scabies (n = 1) and erectile dysfunction (n = 1). Hospital pharmacy (n = 20) was differentiated into subthemes for the production of medicine, hospital pharmacy services, or medication handling. Production of medicine (n = 11) represented projects related to either work environment for staff or Good Manufacturing Practice (GMP). Hospital pharmacy services (n = 2) were about communication related to hospital pharmacy services. The last subtheme, medication handling (n = 7), was related to projects about the dispensing, administration, storage, or other handling of medicine such as hygiene or optimisation.

### 3.1. Challenges in Projects Due to the COVID-19 Pandemic

The COVID-19 pandemic affected how pharmacies could work, since they had to stay open during the lockdown to ensure the safe and effective use of medicine because they are categorised as essential healthcare providers. For the completion of third-year projects, challenges experienced due to the COVID-19 pandemic were investigated from an analytical workshop where supervisors discussed their experience with supervising students throughout the completion of the projects.

The supervisors observed that students’ possibility to collect data was reduced. This was due to much fewer patients visiting the pharmacy. Students’ access to other pharmacy branches was limited because of the need to show consideration for colleagues and patients, which further reduced access to patients. In some cases, there was also limited access to community pharmacy staff. At hospital pharmacies, there was limited access to, e.g., production facilities because of restrictions on the number of nonessential persons who were allowed to gather data at pharmacies and production facilities.

The planned methods to gather data in the projects often had to be amended. Even before starting data collection, students were aware that restrictions were in place and that their initial ideas may have been impossible. Methods such as observations and interviews in person became almost impossible to carry out and had to be replaced by surveys and online interviews. In practice, the number of days when students could collect data from patients and staff, and the time spent observing a process were reduced. Access to external collaborators for interventions (e.g., teaching) and data collection, such as nursing homes, schools, general practices, hospitals, and other community pharmacies was generally limited.

Frustration among students created declining enthusiasm, but also resulted in more constructive reflection and higher satisfaction with supervision. When methods had to be changed, students had to reflect on their choices, use their supervision in a more creative way, find new ways to generate data, and consider how the changes affected the quality of their project.

Supervision became more tailored to meet students’ need for advice and sparring. Scheduled supervision during college hours or emails out of hours were replaced by flexible face-to-face video calls. With increased flexibility, students were able to request supervision when they had the need, resulting in more focused supervision for the students.

The task for supervisors changed during the COVID-19 lockdown. Supervisors spent more time sparring with students, and together they had to be more creative to find solutions. Sparring between supervisors and students increased during the COVID-19 lockdown, and new visions and ideas for future supervision and teaching were generated. The cooperative relationship between community and hospital pharmacies and the supervisors was strengthened as a result of the increased need for communication during the lockdown.

Community and hospital pharmacies have acted very flexibly to accommodate for changes in students’ projects due to COVID-19 restrictions.

### 3.2. Comparing Students’ Results

Statistical analysis of data for 2016–2021 showed that the average grade was best (highest) in spring 2020, with 7.37, and the lowest in spring 2021, with 6.94. Comparing the data for 2020 and 2021 demonstrated no statistical difference between the two years (*p* = 0.09). Comparing the means of grades for all the years demonstrates no statistical difference (*p* = 0.84). For details, see Table 3.

## 4. Discussion

This review of third-year projects by pharmacy technician students included 140 projects allocated to eight overall themes. Descriptive analysis shows that students worked with different themes of pharmacy practice representing all six elective courses focusing on medication safety for patients. Projects from community pharmacies covered many different pharmacy services, such as counselling on different medications and the handling of medicines. From hospital pharmacies, projects particularly focused on handling and producing medicine. A more thorough analysis of the themes of medication counselling in community pharmacy and hospital pharmacy was chosen for a more detailed review, as the themes represented core tasks for pharmacy technicians in either community pharmacies or hospital pharmacies. Despite the challenges to studying and carrying out projects during the COVID-19 pandemic lockdown, students managed to successfully complete their projects with the mean grade for projects at the same level as the previous five years (2016–2020).

On the basis of supervisors’ experiences, students, pharmacies and supervisors had to demonstrate flexibility and learning agility to accommodate the ever-changing COVID-19 restrictions. According to discussions with supervisors, the students experienced limitations in all activities of data collection, making it close to impossible to collect data from interviews in person, observations or to carry out interventions at schools, care homes and general practitioners etc. This fact may come across as frustrating, but the dialogue with supervisors also revealed that the pandemic had facilitated a more complex reflection from both students, supervisors and community and hospital pharmacies, resulting in new and alternative ways to conduct projects.

From the literature, we only managed to identify 1 study reporting pharmacy technician students’ experience with COVID-19 [3], but experiences from pharmacist students’ internships have been reported in detail from Denmark and in more anecdotal terms from several other countries. Danish pharmacist students had the opportunity to focus their internship report for 2020 on their experience with COVID-19. On the basis of reports from 47 students, an overall picture emerged of community pharmacies showing a high level of adaptability. Initially, the practical adaption of the pharmacy to accommodate for the pandemic with separation from patients using plexiglass or face shields, restrictions to limit the number of patients in the pharmacy to increase distance, and logistical challenges to provide alcohol-based hand gel, face masks, and gloves was challenging. The second challenge was to achieve the required changes to provide the expected level of counselling on the safe and efficient use of medicine in the context of COVID-19 and with different levels of anxiety among patients. Pharmacy students reported that pharmacies were very good at adapting to the changes by increasing online sales, counselling over the phone, and introducing drive-by and to-go sales. Pharmacy management was adjusted from value-based management to crisis management to give room for quick changes in structure and adapting new legislation. From the students’ reports, it is clear that the situation has demonstrated to them the role pharmacies have in the healthcare sector with the no-appointment access for patients to get advice from qualified health care professionals [4,5,6].

The challenges and need for adaptability experienced by Danish pharmacy technicians also reflect the experienced challenges by pharmacies providing pharmaceutical care, and have affected the priority for research in pharmacy practice. The pandemic demonstrated the need to accelerate the change in paradigm for pharmacies, from dispensing medicine to patient-centred healthcare in collaboration with the patient and other healthcare professionals [7]. Likewise, the pandemic highlighted the need to prioritise research in pharmacy practice to focus on subjects such as medicines and vaccines, pharmacy services, pharmacy staff workforce and education and training. The change in learning and training experienced by Danish Pharmacy technician students has also been reported from the education of pharmacy students [2].

International references for internships of pharmacy students’ experience with COVID-19 are sparser. Community pharmacy students in the UK doing their final 52-week pharmacy training experienced difficulties with cancelled registration assessment, while students in Nigeria were affected by travelling restrictions and hence the possibility to do their internships [8]. In hospital pharmacies in Taiwan, internships were organised according to three levels of pandemic levels. At low levels, education and internships were carried out as planned; at the highest level of the pandemic, the internship was terminated, and all teaching was conducted on virtual platforms. The higher the pandemic level was, the more restrictions and challenges in the education there were [9].

### Limitations

Experienced challenges due to the COVID-19 pandemic were identified from dialogue with supervisors at an analytical workshop and not students. Entering into a dialogue with students is likely to add additional challenges and how to overcome those to the existing list.

## 5. Conclusions

Although students generally had to overcome more challenges than anyone could have imagined, 178 third-year projects were completed, with 140 students consenting to have their project included in the mapping. The 140 included projects were thematised into eight themes: Advanced pharmacy services, digital patient support, organisation and collaboration, handling of medicine, automated dose dispensing, medication counselling in community pharmacy, hospital pharmacy, and others, covering all six elective courses. The completion of projects was possible in a year with COVID-19 challenges because supervisors, hospital pharmacies, community pharmacies, and in particular the pharmacy technician students themselves demonstrated great flexibility and creativity in doing their third-year projects. Students’ possibilities became limited, as the extent of investigation and data collection were challenged by numerous and ever-changing restrictions due to the COVID-19 pandemic. On the other hand, the many changes forced students, community pharmacies, hospital pharmacies, and supervisors to consider different methods and discuss strengths and weaknesses. Results demonstrate that students graduated with projects covering a broad range of pharmacy practice at a level no different from students completing their projects before the COVID-19 pandemic.

## Figures and Tables

**Figure 1 pharmacy-10-00033-f001:**
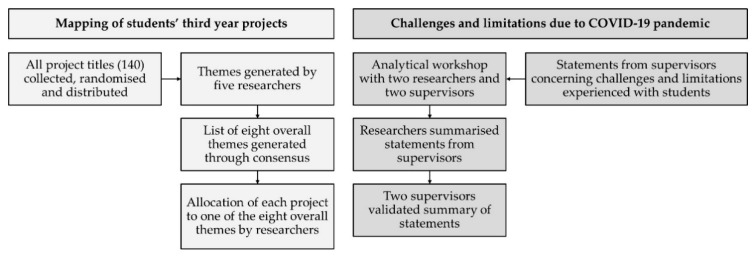
Overview of process of mapping students’ third-year projects, and challenges and limitations due to COVID-19 pandemic.

**Table 1 pharmacy-10-00033-t001:** Elective courses and learning objectives.

Elective Course	Learning Objectives
Clinical pharmacy practice in community pharmacy	Be aware of the balance between clinical responsibility and commercial success. Be aware that community pharmacies are part of the primary health care system. A complex understanding theory and relevance of patient safety and rational pharmacotherapy. Ability to advise patients about all aspects of use of medicine.
Clinical pharmacy practice in hospital pharmacy	A complex understanding of theory and relevance of patient safety and rational pharmacotherapy. Understanding and contributing to hospital pharmacy services.
Production of medicine in hospital pharmacy	Be aware of quality assurance in production. Ability to explain principles for Good Manufacturing Practice (GMP). Ability to explain and handle physical, psychological, chemical, and biological impact in the production of medicine.
Community pharmacy marketing and sales	Ability to plan, carry out, and evaluate marketing campaigns and sale promotion. Conducting and discussing personal sales.
Organisation, management, and collaboration	Understanding the meaning and being able to solve problems in relation to organisation, management, and collaboration at pharmacies and other workplaces.
Health promotion and disease prevention	Evaluating and using methods to develop and implement campaigns for health promotion.

**Table 2 pharmacy-10-00033-t002:** Overview of all eight themes and number of allocated project titles.

Theme	Description of Theme	No. of Projects
Advanced pharmacy services	New Medicine Service or Medication Adherence Service	7
Digital patient support	Use of digital media or platforms to support or inform patients about pharmacy-related knowledge	5
Organisation and collaboration	Projects related to the organisation of the pharmacy, work environment, marketing, sales, campaigns and external activities at nursing homes or residential facilities for mentally or physically challenged adults	31
Handling of medicine	Dispensing medicine, administration of medicine, production (preparation) of medicine in care homes, community pharmacies etc.	19
Automated dose dispensing	The use of automated dose dispensed medicine by patients and not the actual dispensing process	3
*Medication counselling in community pharmacy*	*Counselling and support of patients’ safe and effective use of medicine in community pharmacies for different patient groups*	*52*
*Hospital pharmacy*	*Production of medicine, patient communication and hospital pharmacy services*	*20*
Other	Projects not related to any of the above subjects	3

Final list of themes, description of themes and number of allocated projects. Themes in italics were further differentiated. For the themes medication counselling in community pharmacy and hospital pharmacy, subthemes were allocated.

**Table 3 pharmacy-10-00033-t003:** Data for 2016–2021 (number of students’ mean, median, and variance). ANOVA of means 2016–2021 (*p* = 0.84).

Year	n (Students)	Mean	Median
2016	168	7.08	7
2017	161	7.02	7
2018	218	7.15	7
2019	217	7.09	7
2020	174	7.37	7
2021	181 *	6.94	7

* Grades from three additional projects were included.

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
