# Peer review of "Mapping of Danish Pharmacy Technician Students’ Third-Year Projects in a Year with the COVID-19 Pandemic"

_pharmacy, 2022, doi:10.3390/pharmacy10010033_

Round 1

Reviewer 1 Report

The publication deals with a topic of little usefulness in pharmacy. The information collected is presented with poor statistical analysis. In addition, conclusions are drawn without the support of the researchers. These are only subjective evaluations of the authors, such as:
"Frustration among students created declining enthusiasm, but also resulted in more constructive reflection and higher satisfaction with supervision from supervisors. When methods had to be changed, students had to reflect on their choices, use their supervision
in a more creative way, find new ways to generate data and consider how the changes
affected the quality of their project".

An additional qualitative analysis could be carried out, which for sure  enrich the research and make the topic more meaningful.

Author Response

Dear reviewer One, thank you for your comments and suggestions to make our manuscript more useful. We have below inserted your comment followed by our replies. 

Comment

The information collected is presented with poor statistical analysis. In addition, conclusions are drawn without the support of the researchers. These are only subjective evaluations of the authors, such as:
"Frustration among students created declining enthusiasm, but also resulted in more constructive reflection and higher satisfaction with supervision from supervisors. When methods had to be changed, students had to reflect on their choices, use their supervision in a more creative way, find new ways to generate data and consider how the changes affected the quality of their project".

Reply

Thank you! We have amended the methods to make it clear that evaluations of the supervision are done by researchers based on statements from supervisors. Also, to visualise a figure of the workflow has been submitted

Comment

An additional qualitative analysis could be carried out, which for sure enrich the research and make the topic more meaningful.

Reply

Thanks for your comment. A qualitative analysis of students statements would indeed enrich the study. It is not possible to conduct interviews as students have graduated and now works at pharmacies all over Denmark. We have included this in our limitations.

Bjarke Abrahamsen

Reviewer 2 Report

  1. The manuscript "Mapping of Danish pharmacy technician students’ third-year projects in a year with the COVID-19 pandemic" does provide some results to support the conclusions.
  2. Can the authors provide one or two figures in the manuscript?

Author Response

Dear reviewer Two

Thanks for your comments and suggestions to improve our manuscript. We have below inserted your comments followed by our replies. 

Comment

The manuscript "Mapping of Danish pharmacy technician students’ third-year projects in a year with the COVID-19 pandemic" does provide some results to support the conclusions

Reply

Thank you. We wanted to investigate if students managed to complete their third-year project even in these difficult times, report some of their challenges and at the same time provide a mapping of their projects.

Comment

Can the authors provide one or two figures in the manuscript?

Reply

Thank you for your suggestion. We have in the author group discussed several possibilities to provide a figure. We decided on a figure describing the workflow for both the mapping of students third-year projects and for the COVID-19 challenges and limitations.

Bjarke Abrahamsen

Round 2

Reviewer 2 Report

It is better now.